# Electromechanical Reciprocity Applied to the Sensing Properties of Guided Elastic Wave Transducers

**DOI:** 10.3390/s23010150

**Published:** 2022-12-23

**Authors:** Bernd Köhler, Lars Schubert, Martin Barth, Kazuyuki Nakahata

**Affiliations:** 1Fraunhofer IKTS, Maria-Reiche Str. 2, 01109 Dresden, Germany; 2Graduate School of Science and Engineering, Ehime University, 3 Bunkyo, Matsuyama 790-8577, Japan

**Keywords:** reciprocity, sensor calibration, transfer function, plate waves, pencil-lead break, guided wave sensors

## Abstract

Guided elastic wave (GEW) transducers for structural health monitoring (SHM) can act as transmitters (senders) and receivers (sensors). Their performance in both cases depends on the structure to which they are coupled. Therefore, they must be characterized as system transducer- structure. The characterization of the transducer-structure as transmitter using a Scanning Laser Doppler Vibrometer (SLDV) is straightforward, whereas its characterization as receiver is non-trivial. We propose to exploit electromechanical reciprocity, which is an identity between the transfer functions of electrical-to-mechanical and mechanical-to-electrical conversions. For this purpose, the well-known electromechanical reciprocity theorem was adapted to the following situation: The two reciprocal states are “electrical excitation and detection of the surface velocity at point P” and “mechanical excitation at P and measurement of the electrical quantities”. According to the derived formulas, the quantities on the mechanical and electrical sides must be chosen appropriately to ensure reciprocity as well as that the corresponding transfer functions are equal. We demonstrate the reciprocity with experimental data for correctly chosen transfer functions and show the deviation in reciprocity for a different choice. Furthermore, we propose further applications of electromechanical reciprocity.

## 1. Introduction

The development of new concepts for nondestructive testing (NDT) and structural health monitoring (SHM) has triggered the design and application of alternative types of sensors and transducers. Various piezoelectric transducers have been proposed for structural health monitoring using guided elastic waves. Examples include disks, foils, and fiber transducers. A reliable characterization of these transducers under various application conditions is necessary because transducer performance depends on several factors. This includes the way the transducer is connected to the specimen (e.g., attached or embedded), the type and thickness of the adhesive layer, and the structure to which the transducer is attached. The characterization of the transducer alone is not helpful. The system of the transducer coupled to the object must be characterized as a whole. Analytical and numerical modeling of transducer behavior is helpful, but it also poses challenges. In particular, implementing all essential aspects of a transducer and its interaction with the structure in the model is difficult. Furthermore, reliable input parameters are missing in several cases.

A robust substitute for or supplement to numerical modeling is the precise and reliable measurement of the transducer’s behavior. The transmission properties of a transducer emitting guided waves can be characterized using non-contact 3D-Laser Doppler Vibrometer (LDV) measurements. If the impulse response of the transducer-structure system is measured using the LDV, the signal for an arbitrary electric excitation can be obtained by the convolution of the pulse response with the excitation signal.

Currently, there are no known methods for characterizing transducers as receivers because of a lack of practical and reproducible excitation sources. In the literature, an often-mentioned point is that the transducer’s performance as a receiver equals its performance as a transmitter due to reciprocity. This statement is true only if the corresponding sender and receiver transfer functions are defined appropriately. This subject was discussed in more detail for electroacoustic transducers by Primakoff and Foldy [1,2] in early papers. Partly based on that, several works on the self-calibration of electroacoustic transducers have been published (see, e.g., [3]). More recent work has applied reciprocity to the numerical modeling of transducer behavior [4]. The author demonstrated that by using reciprocity, numerical expenses for characterizing transducer reception behavior can be considerably reduced. The analysis of Foldy and Primakoff is valid for electroacoustic transducers coupled with a fluid. Ultrasonic transducers coupled to a solid through a thin liquid layer could also be included through a combination of electroacoustic transducer reciprocity and mechanical reciprocity of the propagation medium. However, the Foldy–Primakoff paper is not directly applicable to transducers directly coupled with a solid; for example, shear-wave NDT-transducers are coupled via a (highly) viscous layer. Furthermore, almost all piezoelectric transducers are glued to the surface for SHM tasks. Determining the well-defined receive transfer functions of guided-wave transducers directly through experiments is difficult. To the best of our knowledge, only two papers from one group have described such measurements [5,6].

Our aim was to solve the difficulty associated with the experimental determination of sensitivity for NDT and SHM transducers by generalizing the Foldy–Primakoff approach to solids. We could try a straightforward generalization of the work of Foldy and Primakoff. However, we found that starting with the approach introduced by Auld concerning the investigation of the ultrasonic scattering process was easier [7]. Achenbach reviewed recent literature and provided several application examples of this formalism [8]. Most of these studies have focused on finding and illustrating solutions to wave propagation problems, partially involving piezoelectric transducers. However, in [8], no example was given to explain how this formulation can be applied to determine the receiver transfer functions (sensitivities) of elastic wave transducers out of the sending behavior when coupled to a structure. To the best of our knowledge, similar work has not been published previously.

In the present work, we show that a well-defined reciprocity relation can be found, which is adapted to relate Guided Elastic Wave (GEW) transducer sensitivities based on receiving transfer functions to well-defined sender transfer functions in a very general way. The sensitivity is replaced by a measurement of the corresponding sender transfer function and multiplied by a simple factor.

The sensitivity of a transducer can be defined in several ways. In the next section, various definitions suitable for SHM-Lamb wave transducers are discussed, and one is selected for further use in the following sections. Furthermore, a reciprocity relation is derived, and the relationship between the transmission transfer function and sensitivity of the transducer (that is, the receiving transfer function) is obtained. Because the reciprocity relation is an exact equation, it requires no further experimental proof. Nevertheless, in Section 3, we present an experimental test for a simple case to convince readers of the theoretical approach. Section 4 presents the conclusions and possible applications.

The material presented here was also the basis for a previous symposium publication of some of the authors [9]. The present paper exceeds the previous one in several aspects. Especially, the measurement reciprocity (Equation (6)) is now derived which was previously only claimed to be derivable. Furthermore, [9] contains serious errors that are corrected.

## 2. Materials and Methods

### 2.1. Appropriate Definition of Transducer Receive Transfer Function (Sensitivity)

The “sensitivity” parameter of a transducer can be defined in several ways. The most appropriate choice depends on practical questions, such as the ease of its determination and how well it describes the practical operation of the transducer. We list the possible choices and discuss them, focusing on GEW transducers.

In general, the sensitivity of an electromechanical transducer is defined as the ratio of some type of electrical output to the input, which could be external mechanical excitation, or an incoming wave of a given amplitude. The quantities to be used are a matter of definition. The only existing condition is a linear relationship between the quantities. Some appropriate choices are discussed below. 

The measured electrical quantity is usually chosen to be the current flowing through or the voltage drop across a terminating resistor. The resistor can also be 0 Ω (short circuit, no voltage), or infinite (open-circuit; voltage but no current). The choice of mechanical excitation is relatively more complex. First, it is necessary to determine whether an isolated transducer or the system of the transducer coupled to a propagation medium is described. The first choice has the advantage that sensitivity is a property of the sensor alone. However, there is no known way of deriving the receiving transfer properties of the coupled system “transducer attached to the propagation medium” from sensitivity of the pure (uncoupled) transducer. The former is relevant for practical applications. Therefore, we select the second alternative and concentrate on the sensitivity of the transducer glued to a characteristic “object,” for example, transducers coupled to a solid half-space of a given material, or an infinitely extended plate of a given thickness d and known material properties. Because we are interested in SHM sensors, we chose the plate. For simplicity, the plate should be an isotropic and homogeneous material, such that it is characterized only by its longitudinal and shear wave velocities and density. However, this approach can even be applied to general anisotropic and inhomogeneous plates, a half-space, or other geometries.

For the external excitation mechanical quantity, we can choose between (among others)

(a)an incoming monochromatic plane wave (the “plane” is considered in the 2D coordinate system of the plate);(b)an incoming wave train (e.g., a pulse);(c)a wave generated by a point source, such as a point force on the surface or given point displacement at the surface; or(d)a wave excited by a standard transducer at a given position relative to the receiver. The standard transducer itself is excited by a defined electric pulse.

The sensitivity defined with respect to an incoming monochromatic plane wave (a) is a very valuable quantity. For a plate, a series of such sensitivities are indexed by GEW modes. Additionally, the sensitivity of each mode depends on the orientation of the incident wave vector with respect to the transducer orientation. Based on this, the transducer response to an arbitrary monochromatic incoming GEW can be obtained by simple integration over the angular distribution and summation over the GEW modes. However, all monochromatic sensitivities are difficult to obtain experimentally because they essentially involve wave propagation on an infinitely extended plate. 

The choice of pulse option (b) avoids the difficulty of (a) concerning the infinite plate dimensions. However, the source of the pulse and its spectral composition must be specified and experimentally implemented. Specifying the pulse shape limits the application of this approach. 

The definition of transducer sensitivity based on an excitation according to (d) is equivalent to defining a transfer function in a pitch-catch configuration where the “pitcher” is a standard transducer. However, this approach is not general.

Therefore, option (c) remains. For the transfer function (TF) as the transmitter (sender) and receiver (sensor), we have
(1)$$TF^{send}\left(\omega \right)=\frac{mech^{send}\left(\omega \right)}{el^{send}\left(\omega \right)}\;\;\;\;TF^{rec}\left(\omega \right)=\frac{el^{rec}\left(\omega \right)}{mech^{rec}\left(\omega \right)}$$

Here, all mechanical values refer to a fixed point at the surface (P), and the electrical quantities are measured at the transducer cable end. Some degree of freedom is allowed in choosing between voltage and current for a given terminating resistor. Mechanically, the choice of excitation is not fixed; it can be a force or displacement at the plate surface. Reciprocity between transmission and sensing would mean that the transfer functions $TF^{send}$ and $TF^{rec}$ would be equal, or at least remain at a fixed ratio. As we will see in the next subsection, this determines the choice of quantities used for the sensitivity definition.

### 2.2. Reciprocity between Selected Sending and Receiving Sensitivities

To determine the formulation of the reciprocity between the sending and receiving properties of a transducer coupled to a propagation object, we start with the general formulation of the reciprocity for a piezoelectric solid as formulated by Auld [6]. 

The following derivation is valid for any volume V that contains the sample and a transducer coupled to it (Figure 1). We consider two sets of electromechanical field variables within the said volume V under two different conditions: “A” and “B.” Each field is characterized by the stress tensor $\vec{\vec{\sigma }}$, mechanical particle velocity field $\vec{v}$, electric field $\vec{E}$, and magnetic field $\vec{H}$. The external sources are the fields of force density $\vec{f}$ and current density $\vec{j}$. The surface tractions $\vec{t}$ are given by $\vec{t}=\vec{\vec{\sigma }}\cdot \vec{n}$ with the unit normal vector $\vec{n}$ pointing outwards from the volume. The transducer is completely electrically shielded; that is, the electric and magnetic fields are zero everywhere on the surface S, except on S_cable_, where S crosses the electrical connection. All field variables have been defined in the frequency domain. For sake of conciseness, the frequency dependence on ω is not expressed explicitly.

We identify both sets of field variables by their corresponding upper indices, “A” and “B,” respectively. Then, the reciprocity theorem is given by [7]
(2)$$\begin{array}{c} \overset{}{\underset{S=\partial V}{\int }}\left({\vec{t}}^{A}\cdot {\vec{v}}^{B}-{\vec{t}}^{B}\cdot {\vec{v}}^{A}\right)dS+\overset{}{\underset{S}{\int }}\left({\vec{E}}^{B}\times {\vec{H}}^{A}-{\vec{E}}^{A}\times {\vec{H}}^{B}\right)\cdot \vec{n}dS\\ =\overset{}{\underset{V}{\int }}\left({\vec{f}}^{B}\cdot {\vec{v}}^{A}-{\vec{f}}^{A}\cdot {\vec{v}}^{B}+{\vec{j}}^{B}\cdot {\vec{E}}^{A}-{\vec{j}}^{A}\cdot {\vec{E}}^{B}\right)dV \end{array}.$$

We additionally assume that there is no external force or current density in the volume. Therefore, (2) simplifies to
(3)$$\overset{}{\underset{S=\partial V}{\int }}\left({\vec{t}}^{A}\cdot {\vec{v}}^{B}-{\vec{t}}^{B}\cdot {\vec{v}}^{A}\right)dS+\overset{}{\underset{S}{\int }}\left({\vec{E}}^{B}\times {\vec{H}}^{A}-{\vec{E}}^{A}\times {\vec{H}}^{B}\right)\cdot \vec{n}dS=0$$
which is Equation (4) of [7] (surface S has no internal flaws; therefore, the part corresponding to $S_{f}$ in [7] has been omitted in (3). We follow Primakoff [2] in the evaluation of the second integral, which is nonzero only on $S_{cable}$, where it crosses the electrical cable (Equation (16) in [2]), and obtain
(4)$$\overset{}{\underset{S=\partial V}{\int }}\left({\vec{t}}^{A}\cdot {\vec{v}}^{B}-{\vec{t}}^{B}\cdot {\vec{v}}^{A}\right)dS-U^{B}I^{A}+U^{A}I^{B}=0$$
where U denotes the voltage between the inner and outer parts of the coaxial cable, and I is the current to the transducer. 

To specify the two states, we define that in “A,” the surface is free of any mechanical traction; that is, ${\vec{t}}^{A}=0$ for all $\vec{r}\in S$. This implies that there is no source on the mechanical side. This corresponds to pure electrical sources, and we denote this as the transducer transmission (or sending) state. The second state “B” should correspond to the transducer acting as a receiver (sensor). Setting $I^{B}=0$ is a clever choice, as it means that no electrical energy flows into the transducer in state “B.” This corresponds to open-circuit conditions. For the mechanical excitation in state “B,” we assume a point force $\vec{f}$ acting at *P*. The surface traction can be written as
$${\vec{t}}^{B}\left(\vec{r},\omega \right)=\vec{f}\left(\omega \right)\delta^{2}\left(\vec{r}-{\vec{r}}_{0}\right)$$
where the $\delta^{2}$ is the 2D-Dirac delta function defined on surface S, which is assumed to be sufficiently smooth at ${\vec{r}}_{0}$.

If we now set ${\vec{t}}^{A}\equiv 0$, $I^{B}=0$, and the above expression for ${\vec{t}}^{B}$ in (4), the surface integral evaluates to: (5)$${\vec{v}}^{A}\left({\vec{r}}_{0}\right)\cdot \vec{f}+U^{B}I^{A}=0.$$

As the last step, we bring (5) in the form of transfer functions for states A and B. For this, the vector product must be resolved by proper choice of the vector orientations. Choosing the exciting point force $\vec{f}\left(\omega \right)$ to be parallel to the measured surface velocity ${\vec{v}}^{A}\left({\vec{r}}_{0},\omega \right)$ resulting from electrical excitation by $I^{A}$ is not practical because the direction of ${\vec{v}}^{A}\left({\vec{r}}_{0}\right)$ generally depends on frequency (or on time after inverse Fourier transformation), which is not feasible for the excitation force. Therefore, we fixed the orientation of the excitation point force $\vec{f}$ as ${\vec{e}}_{f}=\vec{f}/\left|\vec{f}\right|$ and considered the projection of the velocity ${\vec{v}}^{A}\left({\vec{r}}_{0}\right)$ in its direction as $v^{A}={\vec{v}}^{A}\cdot {\vec{e}}_{f}$. Then, we get
(6)$$\frac{v^{A}}{I^{A}}=-\frac{U^{B}}{f^{B}}$$
with $f^{B}=\left|\vec{f}\right|$.

This means that the ratio of voltage $U^{B}$ generated under open electrical conditions ($I^{B}=0$), owing to point surface force ${\vec{f}}^{B}$ at P (receiving conditions), equals (up to the sign) the ratio of the surface velocity at the same point P, projected onto the orientation of the excitation force, to the current $I^{A}$ flowing into the transducer as the excitation (sending conditions). Equation (6) was given already in [9] as Equation (2), however without deviation and incorrectly without the minus sign. Instead of the external point force $f^{B}$, we used there a “point traction” which is also corrected now. 

In the following, we do not change these two cases A and B. We assume that in state A, the transducer acts as a sender excited by current I(t)*,* with no mechanical excitation, such that all surface tractions are zero. In state B, the transducer acts as a receiver. The current at the electrical connections is zero, but mechanical excitation of the specimen by a surface point force, f(t), is applied. Therefore, we use the upper indices A and send as well as B and rec interchangeably; for example, $I^{A}=I^{send}=I$*,*
$v^{A}=v^{send}=v$.

The left-hand side of (6) has the form of a sending transfer function, as defined in (1), and the right-hand side has the form of the corresponding receiving transfer function. This equation tells us how $tr^{send}\left(\omega \right)$ and $tr^{rec}\left(\omega \right)$ must be defined in (1) (i.e., which mechanical and electrical parameters should be chosen) to ensure that both transfer functions become equal. With
(7)$$tr^{send}\left(\omega \right)=\frac{v^{send}\left(\omega \right)}{I^{send}\left(\omega \right)}\;\;\;\;\;\;\;tr^{rec}\left(\omega \right)=-\frac{U^{rec}\left(\omega \right)}{f^{rec}\left(\omega \right)}$$
we have according to (6)
(8)$$tr^{send}\left(\omega \right)=tr^{rec}\left(\omega \right)$$

Thus, both transfer functions are identical. Because the physical quantities in the sending transfer function (v and I) are different from those in the receiving transfer function (U and f), in the subsequent discussion, the upper indices “send” and “rec” will be dropped for the sake of simplicity. Applying the inverse Fourier transformation to convert to the time domain and using convolution, we obtain
(9)$$U\left(t\right)=-\int tr^{rec}\left(t-\tau \right)\;f\left(\tau \right)d\tau$$
and
(10)$$v\left(t\right)=\int tr^{send}\left(t-\tau \right)\;I\left(\tau \right)d\tau$$
with
(11)$$tr^{send}\left(\tau \right)=tr^{rec}\left(\tau \right)$$

Assuming that the domain type will be clarified by the context and argument, we have used identical symbols for the functions in the time and frequency domains. Using the symbol “*” for convolution, the above expressions can be written as
(12)$$U\left(t\right)=-tr^{rec}*f$$
and
(13)$$v\left(t\right)=tr^{send}*I$$

Both equations refer to the same point ${\vec{r}}_{0}$ at the surface and the same surface unit vector. In (13), v is the surface velocity component of $\vec{v}$, measured in the direction of the surface point force $\vec{f}$. 

The proposed strategy to determine the open-circuit voltage response $U^{rec}\left(t\right)$ of a given transducer-plate system for point excitation at P with force $\vec{f}\left(t\right)$ is as follows:
1.The sending pulse response $tr^{send}\left(\tau \right)$ is determined using (10). This can be achieved by ensuring that the current to the transducer is a δ−pulse, defined as $I^{send}\left(t\right)=I_{0}\delta \left(t-t_{0}\right),$ and measuring the velocity projection $v^{send}\left(t\right)={\vec{v}}^{send}\vec{f}/\left|\vec{f}\right|$ at P. This yields $tr^{send}\left(\tau \right)=v^{send}\left(\tau \right)/I_{0}$. If the current cannot be transmitted as a pulse, a deconvolution must be applied to the signal. 2.The voltage for the receiving case is determined using the reciprocity relation (11), that is, $tr^{rec}\left(\tau \right)=tr^{send}\left(\tau \right)$, and $U\left(t\right)=-tr^{rec}*f$.


The following is a summary of the preconditions assumed when deriving the reciprocity relationships (6) and (11):(a)The system can be described by electrical and mechanical field variables that are coupled by linear and time-invariant constitutive equations. The coefficients in these equations must satisfy certain symmetry conditions, which are typically fulfilled [2].(b)The coupling between the mechanical and electrical quantities is purely electric (piezoelectric and/or electrostatic) or magnetic (magnetostrictive and/or electromagnetic) [2].(c)There is no external electromagnetic field except at the transducer cable end [2].(d)The system is free (no surface traction, volume forces, or external currents) other than at the excitation ports.

Condition (b) is usually fulfilled with NDT and SHM transducers. However, electrical shielding (c) and traction-free support of the system (d) are not always met and must be discussed. Condition (a) excludes nonlinear and hysteretic behaviors. However, heterogeneous and anisotropic materials are not excluded. There are no additional special requirements on the transducer, its coupling to the specimen, on the surface of the specimen or its internal structure.

### 2.3. Specifications of the Hardware 

The transducer used for the experiments was a circular piezoelectric disk with a thickness of 400 µm, a diameter of 10 mm and a free resonance frequency of f=5 MHz. The disk was glued onto a planar aluminum plate with dimensions of 1000 mm × 1000 mm × 2 mm. The distance between the disk center and the excitation/detection point was 150 mm (see Figure 2).

For both the excitation of the transducer in the sending case (left part of Figure 2) and the amplification and digitalization of the received signals, we used an IKTS inhouse development Multi-channel Acoustic measurement System (MAS) [10]. The sampling rate of the MAS was 4.17 MHz per channel for all measurements.

Sensor receiving case: A pencil-lead break (PLB) was used to generate a rather defined excitation. The voltage at the transducer was measured with MAS at amplification of −4dB. The recording was threshold value triggered.

Sending case: A Lecroy WaveRunner 64Xi scope was used for measuring of the actual transmitted voltage. The velocity signals at point P were measured with a Laser doppler vibrometer from Polytec. The velocity decoder output with sensitivity 25 (mm/s)/V was used. This analog output signal was digitized by MAS after 14 dB amplification and a 500 kHz low pass filtering for noise reduction. The noise was further reduced by 256-fold on average. 

Due to the different bandwidths of the devices used, all signals were limited in frequency to build a concerted region of validity. As the PLB excitation reaches frequencies of approximately 1 MHz, our LDV decoder has a low frequency cut-of at 50 kHz and to have additional margin, this region was set at 100 kHz to 500 kHz.

## 3. Results and Discussion

We aim to verify the measurement reciprocity, that is the equality (6) of the transmitting and receiving transfer functions. However, obtaining experimental data for the quantities in (6) is not trivial, particularly for the receiving side ($U^{B}$ and $f^{B}$). Therefore, we derive from Equation (6) a further equation between related but more directly measurable quantities. The fulfillment of the derived Equation (16) (see below) can be interpreted as an indirect confirmation of (6).

We consider a special case in which the vector $\vec{f}$ is normal to the surface; that is, $\vec{f}||\vec{n}$. Therefore, both the excitation force in the transducer receiving case and the determined velocity component in the transducer transmitting case are perpendicular to the surface.

The reciprocity relation is tested in the time domain by proving that (9) and (10) are connected through (11). The simplest method would be to use a δ-function excitation in both cases to reduce the integrals to their kernels. However, experimental data are available only for voltage pulses and step-like force excitations.

### 3.1. Transmission Experiment

In the transmission experiment, the transducer was excited by a short rectangular voltage pulse *U* with a duration of 1 µs. Current $I^{send}\left(t\right)$ was calculated using the electrical impedance of the transducer, low pass filtered for noise removal, and displayed in Figure 3.

The velocity measured by the LDV was band-pass filtered between 100 kHz and 500 kHz according to the concerted region of validity and displayed in the first row of Figure 4.

### 3.2. Reception Experiment

To evaluate the receiving case, the voltage response of the system to surface force excitation must be measured. Generating a pulse-like surface force is difficult. Therefore, we used step-like force excitation using a pencil-lead break (PLB). This method is widely used in acoustic emission testing for transducer coupling control and as a standard excitation source [11]. The PLB produces a step-like force release. For times $t<t_{0}$, the force is constant and directed antiparallel to the normal vector $\vec{n}$ and thus negative $-f_{0}$. After the pencil-lead has broken, the force is zero. This force release can be described using the Heaviside step function *H* as follows:(14)$$f\left(t\right)=-f_{0}H\left(-t+t_{0}\right)$$

The force release has a rise time of approximately 1 µs, corresponding to an upper frequency limit of 1 MHz. The pulse shape of PLBs is reproducible for these frequencies. However, the amplitude $f_{0}$ and time $t_{0}$ may have variations.

Setting (14) with $t_{0}=0$ into (9)
$$U\left(t\right)=f_{0}\overset{\infty }{\underset{t}{\int }}tr^{rec}\left(\tau \right)d\tau \;$$
and differentiating with respect to t gives
(15)$$U\prime \left(t\right)=-f_{0}tr^{rec}\left(t\right).$$

That is, the time derivative of the voltage response due to a PLB step force corresponds to the voltage response due to a pulse force excitation.

The last row of Figure 4 shows the derivative of the voltage after PLB excitation. As it is difficult to reproducibly obtain the excitation magnitude $f_{0}$ of the PLB, it is represented in arbitrary units. 

### 3.3. Verification of Reciprocity

As mentioned earlier, experimental verification of the reciprocity of the measurement provides a validation of the assumptions made in deriving the analytical expression for reciprocity and its mathematical correctness.

Finally, we aim to verify the validity of (11) using the transfer functions given by (12) and (13). The receiver transfer function can be obtained directly using (15). The sender transfer function is relatively more difficult to obtain because the current pulse in our experiment was not delta-like. As the measured velocity is the convolution of the sender transfer function pulse response by the actual current pulse, we could deconvolute the measured velocities. However, deconvolution is prone to noise. Therefore, instead of verifying the correctness of $tr^{rec}=tr^{send}$, we show that its convolution with the actual current pulse $I=I^{send}\left(t\right)$ is fulfilled.
(16)$$v^{calc}∶=I*tr^{rec}=I*tr^{send}=v$$

The velocity measured during the sending experiment is represented in (16) by v. The receiving pulse response $tr^{rec}$ is obtained from (15) up to an unknown strength $f_{0}$. The “calculated velocity,” $v^{calc}$, in (16), must be identical to v if (11) is satisfied.

The upper row of Figure 4 shows the measured out-of-plane velocity v as a response of the system to the excitation of the transducer with the current pulse $I^{send}\left(t\right)$. The graph on the left-hand side shows several pulses. The first two are interpreted as the result of the two lowest-order plate waves, S0 and A0, traveling directly between the transducer and detection point. The time difference between the centers of these two wave packets is estimated to 25 µs. For the travel path of 150 mm, this results in a difference of the slowness $\Delta s=s_{A0}-s_{S0}=1/v_{A0}-1/v_{S0}=\left(1/6\right)µs/\mathrm{mm}$ confirming the interpretation of these pulses. The pulses starting after 300 µs originate obviously from the reflections of the waves at the plate boundaries.

The quantity $v^{calc}:=I*tr^{rec}\sim I*U\prime$ is shown in the middle row. As expected, we see good agreement between v in the upper row and $v^{calc}$ in the middle row. This agreement is particularly evident in the zoomed-in graphs on the right-hand side and in Figure 5, where both quantities are plotted in one graph. 

In Section 2.2, we derived the reciprocal behavior of the transducer-specimen system for a specific quantity at its electrical and mechanical “ports.” To demonstrate that reciprocity does not hold for any choose of quantities, we consider a voltage input at the transducer as the sender. The reciprocity statement could then be defined in the following way: “The response of the system (out of plane velocity at point P) to a voltage pulse at the transducer is equal to the voltage measured at the transducer due to a normal force pulse excitation at point P.” In other words, we define the following: $${\hat{tr}}^{send}\left(\omega \right)=\frac{v^{send}\left(\omega \right)}{U^{send}\left(\omega \right)}$$
instead of (7) and assume ${\hat{tr}}^{send}\left(\omega \right)=tr^{rec}\left(\omega \right)$ or in the time domain ${\hat{tr}}^{send}\left(\tau \right)=tr^{rec}\left(\tau \right)$, respectively. In the following, this will be called “presumed reciprocity.”

The data were obtained using voltage pulse excitation. Therefore, equivalent to (13), we obtain $v\left(t\right)={\hat{tr}}^{send}*U^{send}={\hat{tr}}^{send}$. In other words, we directly obtain the sending transfer function by measuring v(t). The reception experiment is the same as before; therefore, (15) is unchanged. The validity of the “presumed reciprocity” ${\hat{tr}}^{send}\left(\tau \right)=tr^{rec}\left(\tau \right)$ in the system would therefore imply that the proportionality v(t)~U′(t) holds.

The last row in Figure 4 shows U′(t) which must be compared with v(t) shown in the first row. Apparently, the general structures of the wave packets are similar. We observe that the two wave packets (S0 and A0 Lamb mode) travel directly between the transducer and point P, and late arrivals are caused by interference of reflections at the plate boundaries. However, the detailed structure of the direct wave packets is significantly different. This can be interpreted as follows: the correct choice of “termination” at the electrical port influences significantly the detailed structure of the emitted elastic wave pulses and signal shape of the received waves. However, the propagation of elastic waves in the plate away from the transducer is only slightly influenced by the electrical termination. 

Let us return from the “presumed” (proven not to be a correct) to the “correct” reciprocity expressed by Equation (16). In Figure 5, the normalized signals are plotted together over time. They show an evident agreement in the phase. There are still slight deviations in the arrival time differences of both wave packets, indicating (“slight”) violations of the assumptions for the measurement reciprocity. Properties of the plate such as waviness or surface roughness are excluded as reasons, as the object can be of any shape. Only the absence of external surface tractions is required. For the same reason material inhomogeneities and anisotropies cannot lead to reciprocity violations. Moreover, the coupling of the transducer to the specimen is included in the reciprocal system and can be arbitrary. We assume instead that the system is not identical for the transmitting and the receiving experiment. The measurements were not performed immediately after each other. Therefore, temperature or other environmental conditions could have changed between the measurements, which caused a change in the propagation velocities of the different modes between the measurements. In addition, small position differences between the points for velocity detection and force step excitation could have occurred. The last reason is the most probable: the PLB excitation in the receiving experiment was monitored additionally by the LDV, which was kept at an unchanged position compared to the transmission experiment. So, the PLB´s excitation point was nominally the same as the detection point, but in reality, it was slightly off. The assumption of a travel distance increase of only 2 mm in the reception experiment explains the increased time difference of about 0.4 µs between the S0 and A0 wave packets for $v^{calc}$.

We did not compare the amplitudes and neglected the time offsets between $v^{}$ and $v^{calc}$ because we did not know either excitation force of the PLB nor the exact time of the break event. However, this does not mean that reciprocity is invalid for these quantities. It just could not be verified using the available data.

## 4. Summary and Conclusions

Characterizing the sensor properties of electromechanical transducers is essential for GEW applications. Currently, powerful experimental tools to characterize the transmission properties of these transducers are available, such as 3D (Scanning) LDV. With the SLDV, the response of the transducer-specimen system can be obtained for any electrical excitation at any point on the surface of the specimen in the form of a surface displacement vector or surface velocity vector. A prevailing question is whether reciprocity could be used to obtain the receiving properties of a transducer coupled to a structure based on the transmission properties.

Starting with the work of Primakoff and Foldy [2] and using the formulation given by Auld [7], the reciprocity relation (5) for a piezoelectric transducer coupled to a solid was derived. This relationship determines how transfer functions should be defined to ensure that the transmission and reception transfer functions are equal.

Using existing experimental data, the fulfillment of reciprocity in the sending and receiving properties of the sensor-specimen system was verified. However, the reciprocity between sending and receiving transfers may not be satisfied if the corresponding transfer functions are defined arbitrarily. For example, the transfer functions differ considerably if the surface velocity is chosen as a response to a voltage pulse (instead of a current pulse). Thus, the truth of claims (often found in publications) that, in some arrangements, the transmission and reception transfer functions are equal due to reciprocity must always be verified. For reciprocity to be strictly valid, such a statement should be derived from a reciprocity theorem whose underlying assumptions are fulfilled. 

The “presumed reciprocity” was not derived from a valid reciprocity theorem and showed significant deviations in the detailed signal structure; however, the agreement in the overall signal behavior over long periods of time seemed to be sufficient. Apparently, these late time signal parts were strongly determined by wave propagation in the structure and only slightly by the transducer–object interaction. That could be a reason why different kinds of “presumed reciprocity” may work in many cases, even though no strict proof of its validity exists. Examples can be found, especially in wavefield mapping using scanning laser excitation instead of SLDV detection [12,13]. However, to be on the safe side, only proven reciprocity relations should be applied.

To avoid confusion let us state again explicitly the following. When we derive the reciprocal behavior in measurements from a reciprocity theorem, then this reciprocal behavior is always valid whenever the assumptions of the theorem are fulfilled. In our case, reciprocity is not compromised by: heterogeneous materials, any shape of the specimen, including rough surfaces and waviness, as well as perfect or imperfect coupling and any aperture of the transducer. However, the reciprocal behavior in measurements is not guaranteed (and most probably not given) if the assumed options of the reciprocity theorem are not fulfilled, as in the case of hysteresis and nonlinear materials.

This work can be extended from point excitation/detection on the structure to excitation/detection by a second transducer under study. The 3D SLDV allows measurement of the complete vector field of the surface velocity. The reciprocity proved in this study allows the calculation of the response of the system from such measurement data to point forces of arbitrary orientation acting at any point on the surface. A weighted superposition of these solutions can be performed. This implies that any distribution of surface tractions can be assumed, and the response of the transducer to that distribution can be obtained. There are many interesting examples of such approaches. Let us assume that the surface traction distribution generated by a second transducer in the interface is known. Then, the signal received by the first transducer can be calculated owing to the waves sent by the second transducer. This second transducer could be a thin piezoelectric wafer glued to a plate, which is commonly used in SHM with guided elastic waves. Other examples include lightweight fiber-based SHM transducers [14,15,16], SHM transducers introducing shear tractions on the surface [17,18,19,20], and electromagnetic acoustic transducers [21,22], to name just a few. Laser excitation of ultrasound can also be studied in both thermoelastic and ablation regimes. There is an advantage of such an experiment—simulation over a pure numerical approach: all the characteristics in the wave propagation and coupling between propagation medium and first sensor are exactly contained. The challenges of correct material parameters and coupling conditions are circumvented.

## Figures and Tables

**Figure 1 sensors-23-00150-f001:**
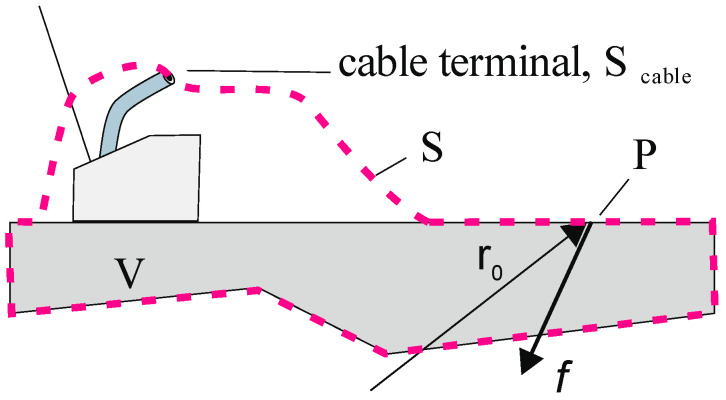
Piezoelectric probe coupled to a solid test specimen; the integration volume V includes the test specimen and the probe and can be of any shape. V has the boundary S indicated by the dashed line. The part of the boundary crossing the electrical coaxial cable is denoted as S_cable_. The point P at the surface has coordinates $r_{0}$ where in state “B” an external point force acts to the surface and the surface velocity is measured in state “A”.

**Figure 2 sensors-23-00150-f002:**
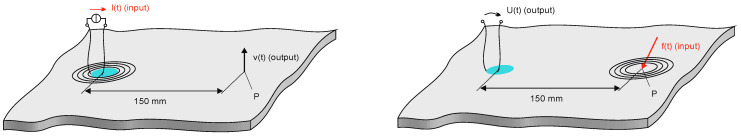
Schematic of experimental setup. (**Left**): the transducer acts as the sender and the out-of-plane velocity is measured. (**Right**): the transducer acts as a receiver, the plate is excited by a point force, and the voltage response under open-circuit conditions is measured. The dimensions of the plate are 1000 mm × 1000 mm × 2 mm.

**Figure 3 sensors-23-00150-f003:**
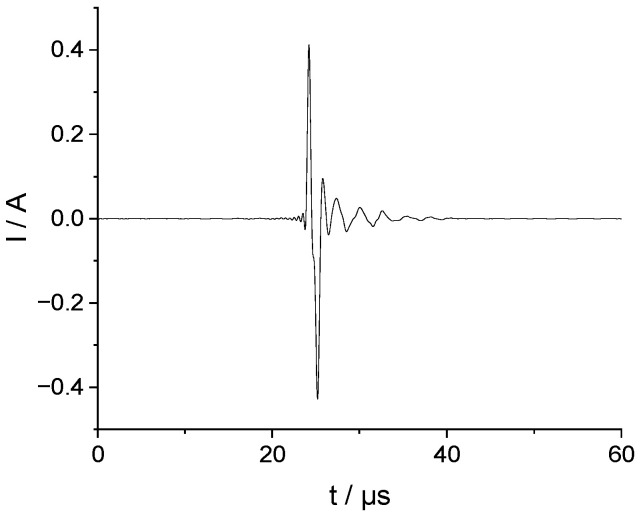
Electric current into the transducer when it acts as sender. The current I(t) was low pass filtered with a cut-off frequency of 2 MHz.

**Figure 4 sensors-23-00150-f004:**
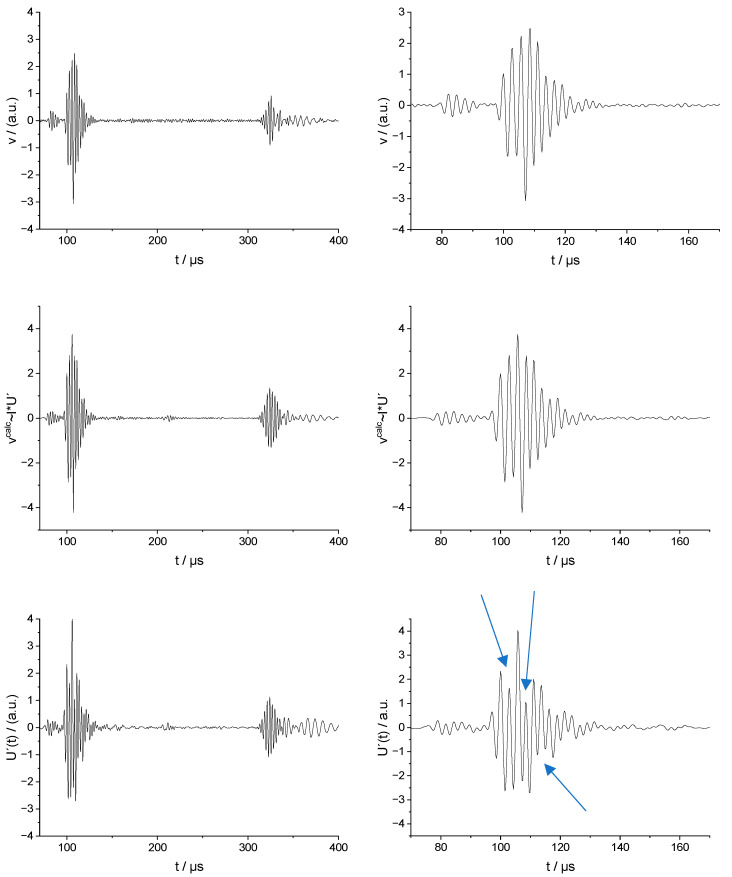
Comparison of the velocity signals. (**Top row**): velocity measured using LDV as response to the current signal $I^{send}\left(t\right)$. **(Middle row**): quantity $v^{calc}=I*tr^{rec}\sim I*U\prime$ and (**Bottom row**): the quantity $v^{calc\_II}=U^{send}*tr^{rec}\sim U\prime$ which must be equal to the measured v when the “presumed” reciprocity would be valid. The arrows indicate positions where $v^{calc\_II}$ significantly deviates from v.

**Figure 5 sensors-23-00150-f005:**
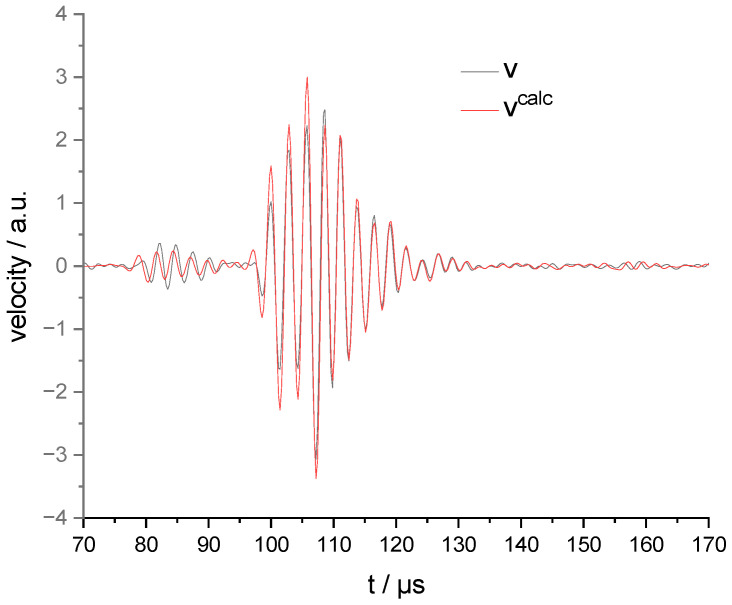
Comparison of the measured surface normal velocity v with transducer as sender and the given (non δ-pulse like) current excitation with $v^{calc}$ calculated according to (16) from the data of the transducer as receiver. Both quantities are shown at arbitrary scales.

## Data Availability

Data is available at https://doi.org/10.6084/m9.figshare.21546318, accessed on 20 December 2022.

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
