# Peer review of "Electromechanical Reciprocity Applied to the Sensing Properties of Guided Elastic Wave Transducers"

_sensors, 2022, doi:10.3390/s23010150_

Round 1
Reviewer 1 Report
1. Title: Please check whether the method is really a sensor characterizion only. The presented methods needs a sending transfer function meaning they only work for transducer that can excite and sense GEW.
2. Introduction l.67 to 80: I would re-structure this paragraph. Start with the literature first and then come to a summarizing paragraph where you state what the novelty of your work is: "We show that a well-defined reciprocity relation can be found, ..."
this leads to an easier transition to section 2 from my point of view.
3. Eq. (4): In which cases does this sign become valid? This influences your statement about the sign in line 214 ff.
4. line 201: Add Eq. or Equation (5)
5. Section 2: Please add hardware specifications on transducers, signal generators, amplifiers, etc.
6. line 276: What kind of piezoelectric disc is used here? Most of the piezoelectric discs induce in-plane strains that are, contrarily to the aforementioned assumption, not normal to the surface.
7. line 302: Check abbreviations for proper introduction on first use, here PLB.
8. line 341: There is no figure 6, supposed to be 5?
9. line 346 and 348: Point A? or Point P?
10. line 349: missing eq. number; denominator: Is this supposed to be send or rec? Pls check.
11. line 359: Usend --> send or rec?
Reviewer 2 Report
Authors of the reviewed manuscript study the reciprocity theorem in application for SHM of elastic plates. Analytically derived relationships can be useful for practice of SHM. They are experimentally validated. The paper is well written, the message and the goal of the paper are clear. By my opinion, this paper would be relevant to the scientific community.
Reviewer 3 Report
Review for Sensors, MDPI
sensors-2061282
Electromechanical Reciprocity Applied to Guided Elastic Wave Sensor Characterization with Laser Vibrometer Measurements
General comments:
The paper deals with electromechanical reciprocity of transducers application to Guided Elastic Wave (GEW) propagation for Structural Health Monitoring (SHM). It consists in a study of the reciprocity principle on a realistic experimental setup.
Useful transfer functions are given:
* The sending signal (electrical to mechanical conversion) is expressed with the velocity to current transfer function.
* The reception signal (mechanical to electrical conversion) is expressed with the force to voltage transfer function.
Specific comments:
* Page 4 of 13, line 175 and 176:
Please replace:
"All field variables have been defined in the frequency domain."
by:
"All field variables have been defined in the frequency domain. For sake of conciseness, the frequency dependence on is not expressed explicitly."
* Page 7 of 13, line 276 and 279:
Please provide some additional information on the experimental setup. The only information we have is:
"The transducer used for the experiment was a circular piezoelectric disk with a thickness of 400 µm and diameter of 10 mm. The disk was glued onto a planar aluminum plate with dimensions of 1000 mm × 1000 mm × 2 mm. The distance between the disk center and the excitation/detection point was 150 mm (Figure 2)."
The center frequency of the piezoelectric transducer is not given, and must be provided. Since this work deals with characterization, it would be very useful to provide the "Electric current into the transducer when it acts as sender" in its international unit, i.e. in Ampere. The electrical impedance of the piezoelectric disk in air and loaded by the aluminum plate would also bring confidence in the given signal processing without any details nor justification:
"The current I was low-pass filtered with a 5 MHz cut-off frequency and the velocity measured by the LDV was band-pass filtered between 100 kHz and 500 kHz."
Why "low-pass [filtering] with a 5 MHz cut-off frequency" the current I?
Why " band-pass [filtering] between 100 kHz and 500 kHz" the current LDV?
* Page 8 of 13, lines 301 and 302:
In the "Reception experiment" paragraph:
"Generating a pulse-like surface force is difficult. Therefore, we used step-like force excitation using a PLB."
What does PLB stand for? Please define the acronym "PLB" at its first occurrence.
* Page 8 of 13, line 306:
The excitation force is defined throughout "the Heaviside step function H", which equals zero for a negative input, and equals one for a positive input. Therefore, the question of the parameters signs between the parenthesis is crucial:
The correct sign inside the Heaviside step function shouldn't it be:
"H( t – t0 )"
instead of:
"H(–t + t0 )"
* Page 8 of 13, line 307:
The excitation shape is not shown to the readers, but it would be useful to give convidence in the given results. The only given comment is:
"The pulse shape of PLBs is reproducible for these frequencies"
The authors must illustrate and/or provide additional information on the pulse shape of PLB.
* Page 9 of 13, line 328:
The equation (16) gives the velocity v as the convolution of the current I by the transfer function trsend. Then, a few lines later (Page 10 of 13, line 349), the estimated transfer function trsend is given as the ratio between of the velocity vsend and the voltage Usend.
The dimensions of those two equation are not coherent and must be corrected.
* Page 11 of 13, line 374 to 379:
In the discussion about the possible imperfections in the superposition of "guessed" and "true" signals from the reciprocity principle, some additional comments are to be provided
* Please discuss the loss of reciprocity implied by the losses: mechanical losses in the plate; mechanical losses in the glue layer; mechanical losses due to the plate waviness and roughness; electrical losses in the transducer.
→ Consequently, please provide additional information on the plate waviness and roughness: waviness and roughness evaluation and/or measurements.
* Consequently, please discuss the possible hysteric effects.
* Consequently, please discuss the homogeneity of the plate.
* Consequently, please provide the full transfer function calculation, including the electrical impedance, the source surface, the glue layer.
* Consequently, please discuss the boundary conditions of the circular source.
* Please provide additional references concerning the loss of reciprocity due to losses.

Round 2
Reviewer 3 Report
All the remarks and comments have be taken into account excpeted the one concerning the Heavyside function parameters.
* Page 8 of 13, line 306:
The excitation force is defined throughout "the Heaviside step function H", which equals zero for a negative input, and equals one for a positive input. Therefore, the question of the parameters signs between the parenthesis is crucial:
The correct sign inside the Heaviside step function shouldn't it be:
"H( t – t0 )"
instead of:
"H(–t + t0 )"
